# 3,4-Dihydro-2(1*H*)-Pyridones as Building Blocks of Synthetic Relevance

**DOI:** 10.3390/molecules27165070

**Published:** 2022-08-09

**Authors:** Sisa Chalán-Gualán, Vida Castro, Ruth Oropeza, Margarita Suárez, Fernando Albericio, Hortensia Rodríguez

**Affiliations:** 1School of Chemical Science and Engineering, Yachay University for Experimental Technology and Research (Yachay Tech), Yachay City of Knowledge, Urcuqui 100119, Ecuador; 2Institute for Research in Biomedicine, Barcelona Science Park, 08028 Barcelona, Spain; 3Laboratorio de Síntesis Orgánica, Facultad de Química, Universidad de La Habana, Ciudad Habana 10400, Cuba; 4CIBER-BBN, Networking Centre of Bioengineering, Biomaterials and Nanomedicine, and Department of Organic Chemistry, University of Barcelona, 08034 Barcelona, Spain; 5Department of Organic Chemistry, University of Barcelona and CIBER-BBN, 08028 Barcelona, Spain; 6School of Chemistry and Physics, University of KwaZulu-Natal, Durban 4001, South Africa

**Keywords:** 3,4-DHPo, 1,4-DHPs, multicomponent reaction, nonconventional synthesis, synthetic precursors

## Abstract

3,4-Dihydro-2(1*H*)-pyridones (**3,4-DHPo**) and their derivatives are privileged structures, which has increased their relevance due to their biological activity in front of a broad range of targets, but especially for their importance as synthetic precursors of a variety of compounds with marked biological activity. Taking into account the large number of contributions published over the years regarding this kind of heterocycle, here, we presented a current view of 3,4-dihydro-2(1*H*)-pyridones (**3,4-DHPo**). The review includes general aspects such as those related to nomenclature, synthesis, and biological activity, but also highlights the importance of **DHPos** as building blocks of other relevant structures. Additional to the conventional multicomponent synthesis of the mentioned heterocycle, nonconventional procedures are revised, demonstrating the increasing efficiency and allowing reactions to be carried out in the absence of the solvent, becoming an important contribution to green chemistry. Biological activities of **3,4-DHPo**, such as vasorelaxant, anti-HIV, antitumor, antibacterial, and antifungal, have demonstrated this heterocycle’s potential in medicinal chemistry.

## 1. Introduction

The study of determinate structures such as small molecules in drug discovery has increased, engaging most in medicinal chemistry. These structures, particularly those based on *N*-heterocycles, represent a class of molecules capable of binding to multiple receptors with high affinity, showing a broad range of biological activity.

In this regard, the 3,4-dihydro-2(1*H*)-pyridones (**3,4-DHPo**) are biologically active *N*-heterocycles analogs of the well-known 1,4-dihydropyridines (**1,4-DHPs**) and dihydropyrimidines (**DHPMs**) [1], which have been introduced in the scientific landscape. In this context, **DHPo’s** as the Milrinone and the Amrinone are drugs with cardiotonic activity successfully used to treat heart failure (Figure 1) [2]. In addition, they and their derivatives have also been reported to possess antitumor [3], antibacterial [4], anti-HIV [5], and other biological activities [6,7,8,9]. These results encourage many research groups to search for potentially active **DHPo’s** analogs.

At the same time, the 3,4-Dihydro-2(1*H*)-pyridones (**3,4-DHPo**) and their derivatives are extensively used as precursors in the synthesis of bioactive molecules such as (±)-Andranginine (Figure 2) [10], elective α1a adrenergic receptors [11], Rho-kinase inhibitors [12], and P2X_7_ receptor antagonists [13].

The synthesis of **3,4-DHPo** derivatives has been important with the passing of time. Different synthetic procedures to synthesize them have taken place in the last years, although the most reported uses are the multicomponent reaction (MCR-4CR) of Meldrum’s acid (**1**), a *β*-keto-ester derivative (**2**), and aromatic aldehydes (**3**) in the presence of ammonium acetate. Additionally, a broad range of techniques have been covered, from the conventional synthesis, through energy resources such as microwave, ultrasound, and infrared assisted to solid and liquid phase synthesis, and recently the chemoenzymatic assisted methodology applied to their asymmetric synthesis (Figure 1).

Due to the importance of this *N*-heterocycle, this review aims to present the principal characteristics of the **3,4-DHPo** structures, the different procedures to synthesize them developed over time, the main biological application, and their crucial importance as precursors to molecules with high relevance in medicinal chemistry. A critical overview of the advantages and disadvantages of the conventional and nonconventional synthetic methods to prepare **3,4-DHPo** are highlighted.

## 2. Nomenclature, Structure, and General Synthesis

The heterocyclic 3,4-dihydro-2(1*H*)-pyridones (**3,4-DHPo**) (Figure 3) are commonly known 4-aryl substituted 5-alkoxycarbonyl-6-methyl-3,4-dihydropyridones (Figure 3a), although the correct IUPAC name is 4-arylsubstituted-2-methyl-6-oxo-1,4,5,6-tetrahydro-3-pyridinecarboxylates (Figure 3b). The calculated tridimensional structure (Figure 3c) shows the atoms’ corroborated disposition in the **DHPo** [14,15,16].

The effort to synthesize **1,4-DHP** derivatives by modifying its ring gave rise to the first accidental synthesis of **3,4-DHPo**, as novel heterocycles that showed interesting biological activity [17]. The general synthesis of **3,4-DHPo** as a racemic mixture involves a multicomponent reaction (MCR 4-CR) of Meldrum’s acid (**1**), a *β*-keto-ester derivative (**2**), and an aromatic aldehyde (**3**) in the presence of ammonium acetate and with or without solvent (Figure 2). Responsible for blocking the formation of the 1,4-dihydropyridines ring is the acidic character of Meldrum’s acid (pKa = 9.97), which is higher than the *β*-keto-ester (pKa = 11.0) [18].

These kinds of molecules present a stereogenic center at C-4. Its absolute configuration (*R*- versus *S*-enantiomer) was a critical factor for the biological activity as an antagonist or agonist of calcium ions [19]. The mechanism reported in Figure 3 has been demonstrated in synthesizing intermediaries and their characterization [20].

The generally accepted mechanism follows a Hantzsch-like pathway with the previous formation of two intermediates; four from Knoevenagel condensation of Meldrum’s acid (**1**) with the corresponding aromatic aldehyde (**3**), and **five** from the reaction involving β-keto-ester (**2**) and ammonia. A Michael-type addition of the enamine **5** on ylidene compound **5** gives rise to **DHPo’s** (Figure 3) [20].

## 3. Synthetic Strategies

Different strategies have been reported to obtain **3,4-DHPo**, conventional synthesis with or without solvent, using different energy resources such as microwave, ultrasound, and infrared to improve the yield and develop green chemistry reactions. Solid-phase organic synthesis (SPOS) and liquid-phase synthesis (LPS) on soluble polymers as supports have also been developed. Finally, chemoenzymatic to asymmetric synthesis has also been established.

### 3.1. Conventional Synthesis

The first strategy used to obtain the **3,4-DHPo** involved the Multicomponent Reaction (MCR) using equimolecular amounts of starting compounds under ethanol reflux for six hours, allowing the synthesis of 3,4-DHPo derivatives with a 15–26% yield [21]. One year later [22], the synthesis of two **3-CN-DHPo** with 51–70% yield from pyran derivatives previously synthesized was reported.

The Suárez group has improved the efficiency of this MCR procedure, replaced the ethanol with acetic acid, and allowed moderate to high yields (Figure 3) [17,18,19,23,24,25,26,27,28,29]. The best results are due to the catalytic effect and the higher boiling point of acetic acid, which improve the decarboxylation step and increase the yields compared with previously reported methods [21,30].

In 2011, Sun et al. developed a protocol to synthesize structurally diverse **3,4-DHPo** via MCR, similar to those previously described by the Suárez group but using arylamines, acetylenedicarboxylate, aromatic aldehydes, and Meldrum acid as starting reagents. The proposed mechanism involves a Michael addition of the enamino ester formed in situ from the reaction of arylamine with dimethyl acetylenedicarboxylate to arylidine cyclic 1,3-diketones [31].

An efficient one-pot synthesis of polysubstituted dihydropyridones derivatives was reported by Khazaei and Anary-Abbasinejad [32]. The reaction was achieved using cyanoacetamide (**6**), aryl aldehydes derivatives (**3**), ethyl acetoacetate (**2**), and ammonium acetate and using pyridine as the catalyst, in ethanol as solvent, and under reflux conditions obtaining 54–68% of yields (Figure 4). The advantage of this method is the use of neutral conditions and the facility of mixing reagents without any previous activation or modification.

An improvement of this method was reported by Dehghan et al. They used similar general conditions, introducing some variation [33]. The researchers changed ammonium acetate to ammonium carbonate and did not use pyridine. Additionally, they also used and compared ethanol and water as reaction solvents. A significant yield improvement was obtained in water (90–96%) compared with ethanol (55–75%).

Furthermore, the Hakimi research group reported a new one-pot, four-component synthesis of 3,4-dihydro-2-pyridone derivatives (3,4-DHPo). The reaction of Meldrum’s acid (**1**), methyl acetoacetate (**2**), benzaldehyde derivatives (**3**), and ammonium acetate and using SiO_2_-Pr-SO_3_H as an efficient catalyst under solvent-free conditions was reported (Figure 5) [34]. The advantages of this methodology are high product yields (78–93%), being environmentally benign, short reaction times, and easy handling.

Moreover, the preparation of 3,4-dihydro-2H-chromeno[4,3-b]pyridine-2,5(1*H*)-dione derivatives using 4-hydroxycoumarin, an aromatic aldehyde, ammonia, and Meldrum’s acid under refluxing with 1-propanol has also been published [35].

Li et al. reported the synthesis of **1,4 DHPs** derivatives with the precursor ethyl 4,4,4-trifluoro-3-oxobutanoate and a short report with the brominating of **3,4 DHPo** using *N*-bromosuccinimide [36,37]. The reaction of α-hydroxyketene-(*S*,*S*)-acetals and active methylenes to obtain 3,4 DHPo derivatives was also reported [38]. Razdan and coworkers reported the synthesis of 3,4-dihydro-2-pyridones, using Bi(III) nitrate immobilized on neutral alumina as the catalyst, in the presence of co-catalyst of Zn (II) chloride with 79–88% yield [39].

On the other hand, fluorinated **DHPo** derivatives synthesis has also been developed. For example, Song et al. reported the synthesis of 3-aryl-4-unsubstituted-6-CF_3_-pyridin-2-ones and ethyl 2-hydroxy-6-oxo-4-aryl-2-(trifluoromethyl)-piperidine-3-carboxylate as essential building blocks for the construction of trifluoromethylated heterocycles, and studying the effect of base and solvents in the reaction obtained 0–93% of yields [40,41]. Further, Smits et al. published the formation of fluorous 3,4-dihydro-2(1*H*)-pyridone-5-carboxylate as a cationic amphiphile. The **3,4-DHPo** moiety plays a key role as a scaffold for attaching cationic head groups [9].

Instead, Dostanic et al. published about synthesizing (substituted phenylazo)-pyridones in the presence of KOH and acetone to obtain 11–61% yield [42]. These were used dyeing polyester fabrics as yellow dyes. Another report described the synthesis of 3,4-dihydropyridones derivatives in the presence of Cs_2_CO_3_ and toluene with 53–66% yield [43]. The synthesis of aza- analogs of **3,4-DHPo** with anticancer activity was reported by Bariwal et al. using benzoylacetone, substituted aldehyde, urea or thiourea with HCl, and ethanol as a solvent with 55–77% yield [44].

Besides, the conventional reaction has been improved by using different catalysts. Zhiqiang et al. reported a three-component cascade reaction to achieve **3,4-DHPo** derivatives using imidazole as a catalyst with ethylene glycol as solvent [45]. Later, Bhattacharyya et al. reported a greener method to obtain **3,4-DHPo** derivatives using a one-pot multicomponent reaction in aqueous media catalyzed by nanostructured ZnO [46]. Additionally, Khazaei et al. used ZnO nanoparticles to give **3,4-DHPo** derivatives under ethanol as solvent [44]. Ziarani et al. published the synthesis of **3,4-DHPo** derivatives by sulphonic-acid-functionalized ordered nanoporous SBA-15 as a nano heterogeneous catalyst via one-pot, four-component reaction under solvent-free conditions [47]. Further, Pradhan et al. presented green protocols to achieve **3,4-DHPo** derivatives using two catalysts such as the vitamin B1 or PEG–SO_3_H in water as solvent [48]. All these described reactions showed moderate to good yields. Besides, those reactions where a catalyst was attached to solid or polymeric supports showed better results due to the possibility of the most efficient purification procedures.

Zhang et al. reported the synthesis of 5-cyano-2-pyridinone catalyzed by Zn-SSA. The silica sulfuric acid (SSA) was modified with zinc chloride to form the novel catalyst (Zn-SSA), which improved the chemo-selectivity in the reaction [49]. The synthesis was developed using 3-dicarbonyl compounds (**7**), malononitrile (**8**), arylaldehyde (**3**), and solvent-free conditions (Figure 6).

The synthesis of 3,4-dihydropyridine-2(1*H*)-ones catalyzed by ZnBr_2_, FeCl_3_, AlCl_3_, BF_3_, Cu(OTf)_2_, In(OTf)_2_, and BF_3_ OEt_2_ was reported by [50]. This method was developed via Blaise reaction forming a cyclic intermediate (**9**) from benzonitrile (**10**) and Reformatsky reagents, which was generated in situ from ethyl bromoacetate (**11**) and zinc power in ethyl acrylate (**12**) and tetrahydrofuran with 0–81% yield (Figure 7).

Paravidino et al. developed a novel four-multicomponent reaction (4CR) of phosphonate (**13**), aldehydes derivatives (**14**), nitriles (**15**), and α-acidic isonitriles (**16**) to obtain **3,4-DHPo** derivatives in 53–88% of yield and with complete diastereoselectivity in favor of the cis-diastereomer (Figure 8) [51,52].

Another report showed the synthesis of **3,4-DHPo** derivatives via tandem olefins isomerization–RCM reaction, through the in situ generated intermediate **18** from readily available N-Allyl amines type **17** as dienes catalyzed by second-generation Grubbs catalyst (ruthenium catalysts) and heated to obtain **3,4-DHPo** derivatives with 57–85% yield (Figure 9) [53].

The conventional synthesis has been offered a broad possibility to obtain the desired product; however, nonconventional energy sources as microwave, infrared, and ultraviolet have also been incorporated to improve the efficiency of this MCR.

### 3.2. Nonconventional SYNTHESIS

-
**
*Microwave-Assisted Synthesis*
**


In the last decades, microwave-assisted organic synthesis has been used as a tool for many known and new organic reactions. In general, its application allowed to reduce reaction times; increase efficiency; selective heating; excellent reproducibility; and, in some cases, avoid or minimize the use of solvents, contributing to green procedure development [54].

In 2003, our group reported the first solvent-free and accessible one-pot condensation reaction of Meldrum’s acid (**1**) in the presence of methyl acetate (**2**), aldehyde derivatives (**3**), and ammonium acetate to obtained 4-aryl substituted 5-alkoxycarbonyl- 6-methyl-3,4-dihydropyridones (**3,4-DHPo**) [20]. The mixtures were irradiated at controlled temperatures and times with continuous mechanical stirring, which provided a good homogeneity of materials and 81–91% yields (Figure 10).

Afterward, another author reported the obtention of **3,4-DHPos** derivatives with a similar technique and 70–92% yields [55]. Jaques et al. reported the quantitative MW-assisted synthesis of 3,4-dihydro-2(1*H*)-pyridones without solvents but in the presence of solid support as a catalyst [56].

Besides, Hernandez et al. published the oxidation reaction of 4*H*-pyrans derivatives (**19**) to obtain 3-cyano-2-pyridones (**20**) in ethanol, using H_2_SO_4_ catalyst source and MW irradiation (Figure 11) [57]. These compounds are hybrid milrinone–enifedipine analogs.

This research group compared the different energy sources for oxidation; carried out the reaction at room temperature, at ethanol reflux, under infrared and microwave irradiation; and obtained 8, 72, 80, and 86% yields, respectively. Furthermore, the reaction times decreased from seven hours at room temperature until seven and five minutes for IR and MW irradiation, respectively. These comparisons show the importance of using different energy resources and their potential. Besides, ultrasound has been the other nonconventional energy source used to efficiently synthesize **3,4-DHPo**.

-
**
*Ultrasound-Assisted Synthesis*
**


The ultrasonic activation is based on cavitation effects, allowing this technique to improve the mass transfer in several organic reactions reported. In 2011, our group published the synthesis of 4-aryl 3,4-dihydropyridone derivatives (**3,4-DHPo**) by ultrasound-assisted technique, through the one-step condensation of Meldrum’s acid (**1**), alkyl acetoacetates (**2**), aromatic aldehydes (**3**), and ammonium acetate, using glacial acetic acid as solvent, at room temperature, and obtained high yields (Figure 12) [58].

The main advantages of ultrasound-assisted synthesis compared with conventional procedures are the milder conditions, the shorter reaction times, and the higher yields that improve the efficiency of the organic synthesis of these heterocycles. In addition, another energy source, such as the infrared-assisted technique, has also been explored.

-
**
*Infrared-Assisted Synthesis*
**


Parallel to the MW-assisted synthesis design of **3,4-DHPo** derivatives, our group also reported the preparation of 3,4-Dihydro-2(1*H*)-pyridones derivatives (**3,4-DHPo**) by infrared-assisted method of the same multicomponent reaction under solvent-free conditions and similar reagents with moderate yields (Figure 13) [59].

To summarize, different energy sources have been used to prepare **3,4-DHPo**, allowing a broad range of products with varying substituent patterns. Particularly, nonconventional techniques such as MW and IR lead to high yields, short reaction times, and safe and straightforward work-up, constituting a notable improvement and involving green chemistry to synthesize these organic molecules. The support (insoluble or soluble)-assisted synthesis of **3,4-DHPo** has also been a successful development.

### 3.3. Solid-Phase Organic Synthesis (SPOS)

One of the most commonly used techniques in combinatorial chemistry is Solid-Phase Organic Synthesis (SPOS), because it allows the rapid synthesis of many structurally diverse molecules in a short time. In 2006, our group published the first SPOS of **3,4-DHPo** derivatives following a solid-support assisted synthetic strategy (Figure 14) [60]. The immobilized acetoacetate (**21**) was obtained by reaction of 2,2,6-trimethyl-1,3-dioxin-4-one (**1**) and Wang resin (0.92 mmol OH/g); the further reaction of **21** in the presence of NH_4_OAc and HOAc led to the corresponding immobilized enamine (**22**), which reacted with the Knovenagel derivatives (**23**) to afford the expected immobilized **3,4-DHPo** (**24**). The heterocycle was cleaved from the resin with 71–85% overall yield (Figure 14) [60].

This technique has been used in response to the increment of target molecules synthesis to combinatorial chemistry. SPOS of **3,4-DHPo** derivatives presents good results and opens the way to synthesize other molecules with biological activity. Additionally, the synthesis of **3,4-DHPo** derivatives using soluble polymers as support has also been studied [61].

### 3.4. Liquid-Phase Organic Synthesis (LPOS)

The employment of soluble polymers as supports in organic synthesis is known as liquid-phase synthesis (LPS). In 2008, Fu et al. reported the LPS of 4-substituted-5-methoxycarbonyl-6-methyl-3,4-dihydropyridones on polyethylene glycol (PEG) 4000 assisted by MW irradiation (Figure 15) [62]. First, the acetoacetylation of PEG was realized to obtain the immobilized acetoacetate (**21**); further, condensation of **21** with aldehyde derivative (**3**), Meldrum’s acid in the presence of ammonium acetate, and solvent-free assisted by microwave irradiation allowed to obtain the immobilized 3,4-DHPo (**24**). The target compound **3,4-DHPo** was obtained after cleavage using NaOMe in MeOH with 88–95% yield.

The LPS of **3,4-DHPo** derivatives showed excellent results and improved the overall results with the solid phase, allowing the one-step condensation, which was not possible in the SPOS procedure. On the other hand, all synthesized **3,4-DHPo** showed at least one chiral center at C4, and all previously reported procedures allowed to obtain the corresponding racemic mixtures. Hence, efforts have been made to search for the chemo-selective synthesis of these derivatives.

### 3.5. Asymmetric Synthesis

The **3,4-DHPo** structure is closely related to the configuration of its chiral center at C4, bringing biological activity. The chemoenzymatic synthesis can search the specific chiral center configuration and, at the same time, use an ecofriendly procedure. Torres et al. reported the chemoenzymatic preparation of a series of racemic 4-aryl-5-(tert-butoxycarbonyl)-6-methyl-3,4-dihydro-2(1*H*)-pyridones (**25**) using several combinations of lipases (PSL, CRL, CAL-A, and CAL-B) and organic solvents such as 1,4-dioxane, DIPE, and TMBE [63]. The authors improved the enzymatic hydrolysis of *R*-diesters (**27**) derivatives, making subsequent S-enantiomer (**26**) separation viable by acid–base extraction procedures (Figure 16). An improvement to this methodology was made by the chemoenzymatic preparation of optically active phenolic 3,4-dihydropyridin-2-ones [64].

Another enzymatic multicomponent reaction (EMCR) was reported using benzaldehyde, cyanoacetamide, ketone, and Acylaze Amano (AA)-catalyzed. This single enzymatic catalyzed reaction is attractive due to its high atom economy, easy work-up process, tolerance to a wide range of substituted reagents (benzaldehyde and ketones), and all mild conditions [65]. The best result was shown in the enzymatic hydrolysis of 4-aryl-5-(tert-butoxycarbonyl)-6-methyl-3,4-dihydro-2(1*H*)-pyridones with CAL-B enzyme and TMBE as a solvent to obtain high ee (93–95%) and moderate yields (30–31%) of (S)-derivative (**26**).

On the other side, Huang et al. reported the first asymmetric synthesis of **3,4-DHPo** derivatives [66]. The formation of monoacid (*R*)-**30** was carried out by desymmetrization or asymmetric methanolysis of prochiral anhydride (**29**) using the organocatalyst **28**, 2-Me-THF, and MeOH as solvents (Figure 17). The conversion was 100%, and the best-observed ee of 80%. The next step was the selective formylation to obtain the intermediate **31**, and finally, the cyclization of **31** with ammonium acetate using acetic acid achieved the final product (R)-**32**, with an overall yield of three steps 48% and >95% *ee* (Figure 17) [66]. The same group also reported the pilot-scale enantioselective synthesis of **32** and kilogram-scale production of *N*-methyl derivative of **32** in an excellent overall yield of ~22% with excellent stereochemical purity (97% *ee*) [13].

Wanner et al. published the enantioselective synthesis of (*R*)-**3,4-DHPo** through *N*-Heterocyclic carbene (NHC)-catalyzed aza-Claisen reaction of enal (**35**) and enamine (**36**) in the presence of N-mesityl catalyst (**33**) and oxidant (**34**); the better bases were DBU, NMM, and i-Pr_2_Net, showing higher enantioselectivity with 60% to quantitative yields, and 79–96% enantiomeric excess (Figure 18) [13].

Vellalath et al. described the enantioselective nucleophile catalyzed Michael/proton transfer/lactamization cascade with 3,4-difluorocinnamoyl chloride (**37**) and the enamine (**39**) in the presence of **37** as a catalyst with a nonpolar solvent and LiCl as an additive, which affected enantioselectivity; this mild process delivered 3,4-DHPo derivative in 78% yield and 92% *ee* (Figure 19) [67].

As time went on, different strategies were developed to obtain **3,4-DHPo** derivatives in higher yields. Using nonconventional sources such as microwave and infrared radiation increased the yield; reduced the reaction time; and, in many cases, avoided the use of solvent, approaching green chemistry. However, access to the equipment can become a handicap. On the other hand, the main disadvantage of described conventional and nonconventional methodologies is the lack of stereoselectivity, which only shows in the asymmetrical synthesis. Nevertheless, a broad range of procedures described shows the importance of these heterocycles, not only because of their biological activity but also because they are crucial starting materials for synthesizing other more complex entities.

## 4. Structural Characterization

The structure of 3,4-Dihydro-2(1*H*)-pyridones has been determined for physical and analytical techniques such as melting point (Table 1), ^1^H- and ^13^C-NMR spectroscopy, mass spectrometry (electron impact (EI) and electrospray ionization (ESI)), and X-ray. These techniques allowed to obtain a broad range of databases of physical properties to synthesize **3,4-DHPo** [20,28,59].

The technique most used in the characterization of **3,4-DHPo** derivatives has been ^1^H-NMR spectroscopy, allowing us to corroborate the ring formation through the ABX pattern, as was explained by our group [68], which showed the protons H1, H3a, H3b, and H4 of the heterocycle ring (Table 2 and Figure 4).

Electron impact (EI) and electrospray ionization (ESI) have also been used to characterize the **3,4-DHPo**. Our group reported the fragmentation patterns of the even-electron ions formed under ESI conditions and the odd-electron ions generated under EI conditions from substituted **3,4-DHPo** [20]. The characteristic fragmentation pattern under EI conditions was also established (Figure 20) [65].

Under ESI conditions, molecular ions [M + H]^+^ and [M-H]^−^ were observed, corresponding with the positive and negative modes. Additionally, some structures were proposed to explain the fragment ions found in the spectra (Figure 21 and Figure 22).

Different authors have confirmed the structure of **3,4-DHPo** with different substituents by X-ray diffraction [17,29,30,69,70,71,72,73], semiempirical (AM1) calculations [25], NMR spectroscopic including NOE experiments, and coupling constants to determinate the structural conformation in solution [17].

There are some structural requirements to the **3,4-DHPo** derivatives conformation, such as the absolute configuration at C-4 (*R*-versus *S*-enantiomer) acting as a molecular switch, the substituted phenyl ring occupies an axial position perpendicularly bisecting the boatlike **DHPo** ring in a synperiplanar orientation, and the *cis*-carbonyl ester orientation concerning the olefinic double bond [17]. The semiempirical (AM1) calculations and NOE experiments defined two conformational structures; a first presents the aryl substituent at C-4 extended in a pseudoaxial position, and a second in which the aryl substituent is in a pseudoequatorial position; besides, the first conformation was 2–4 kcal/mol more stable than second structure. In both arrangements, the pyridone ring presented a twisted boat [17,19,23,24]. The X-ray studies confirmed the pseudoaxial disposition (Figure 5), which was stable in a solid state after the crystallization of ethanol [17].

## 5. 3,4-Dihydro-2(1*H*)-Pyridones (3,4-DHPo) as Synthetic Precursors

Over time, the **1,4-DHP** and **3,4-DHPo** cores have served as scaffolds for the relevant design of more complex entities with various biological activities.

The effort to improve the synthesis of **1,4-DHP** with a broad range of substitution patterns gave rise to 3,4-DHPo as crucial intermediaries. For example, **3,4-DHPo** were converted to the aromatics 4-(3-nitrophenyl) pyridines (Figure 6a) [74], and to alkyl 4-aryl-6-chloro-5-formyl-2-methyl-1,4-dihydropyridine-3-carboxylate derivatives (Figure 6b) by reaction with the Vilsmeier–Haack reagent. **3,4-DHPo** and 6-chloro-5-formyl-1,4-**DHPs** (Figure 6b) have become versatile intermediaries of other compounds (Figure 6) [26,27,28,75,76,77,78,79,80,81].

Starting from **b**, more complex structures have been synthesized. The 1,5-benzodiazepine fused to a dihydropyridine moiety (Figure 6c) have been prepared, and one derivative (**JM-20**, Figure 6c) has shown promising neuroprotective and antioxidant properties [82,83,84,85]. Besides, [3,4-b]pyridines derivatives (Figure 6d) were obtained by treatment of **b** with hydrazine hydrate. Fulleropyrrolidines endowed with chlorine-containing biologically active 1,4-dihydropyridines (**1,4-DHPs**) (Figure 6e) were also prepared using Prato’s procedure [75,80]. The chloro-formyl derivative b also allowed to obtain 1,4-dihydropyridines (1,4-DHPs) bearing a semicarbazone moiety on C5 (Figure 6f) [86] and iminium salts of dihydropyrido[3,2-e] [1,3]thiazines (Figure 6g) [27].

**3,4-DHPo** has been used as an intermediate for the formation of β-lactams derivatives through photochemical cycloaddition (Figure 6h) [87]. Besides, this heterocycle has been incorporated in the diastereoselective synthesis of 3-Oxo-14,15-dihydroandranginine, a unique indole alkaloid with an unusual ring system that includes a tetrahydroazepine unit condensed with an hexahydroquinoline entity in a *trans*–*trans* fashion (Figure 6i) [34]. The α_1a_ receptor antagonist from **3,4-DHPo** was also synthesized, and its efficacy was demonstrated in a screen of prostate contraction model in rats (Figure 6j) [11].

Indazole amide (Figure 6k) [11] was also prepared through saponification of the ester provided by **3,4-DHPo**, which smoothly combined with an indazole to give close pyridine analogs derivatives with different aryl groups. Thus, imidazole amide is an interesting structure with a selective ROCK1 inhibition. Moreover, a series of isoxazolo[5,4-b]pyridin-6(7H)-ones (Figure 6l) [19] have been synthesized by the reaction of novel 3,4-dihydro-2(1*H*)-pyridones with hydroxylamine hydrochloride and following 5-endo-trig cyclization. Additionally, 3,4-DHPos is an intermediate for the synthesis of Furo[3,4-b]-2(1H)-pyridones (Figure 6m) [17,25,88], which act as potential Calcium channel modulators.

Sadhu et al. report an efficient way for the photochemical dehydrogenation of various substituted **3,4-DHPo** to obtain 2-Pyridone derivatives in excellent yields (83–97%) using different photoinduced electron transfer (PET) sensitizers (Figure 23) [89].

Lawesson’s reagent has been widely used as a powerful, mild, and versatile reagent for transforming carbonyl functionalities into their thio analogs. Our group reported the synthesis of 4-aryl substituted alkyl 2-methyl-6-thioxo-1,4,5,6-tetrahydropyridine-3-carboxylates (**40**) with a 29–93% yield. The thionation of **3,4-DHPo** was carried out in a one-step procedure of the **3,4-DHPo** derivatives (**41**) by exposure to microwave irradiation under solvent and solvent-free conditions (Figure 24) [90].

In 2011, our group reported the preparation of *N*-heterocycles using nonconventional synthesis as an ecofriendly approach to producing heterocyclic nitrogen compounds starting with **DHPo**. MW-assisted synthesis (MWAS) of alkyl 4-arylsubstituted-6-chloro-5-formyl-2-methyl-1,4-dihydropyridine-3-carboxylates (**43**) and 4-arylsubstituted-4,7-dihydrofuro[3,4-*b*]pyridine-2,5(1*H*,3*H*)-diones (**44**) from **3,4-DHPo’s** (**42**) were reported (Figure 7) [91]. US-assisted synthesis (USAS) was also used to obtain chloro-formyl derivatives from **3,4-DHPo’s** as starting materials (Figure 7) [27]. MWAS and USAS showed higher yields in shorter reaction times and milder conditions.

Ueyama et al. identified the **3,4-DHPo** derivative **45** as a degradation product of Azelnidipine (**46**) after radical initiator-based oxidative conditions (Figure 8) [92].

**3,4-DHPo** has been used as an enamine precursor to obtain nitrogen heterocycles such as dihydropyridinones by applying the nucleophile-catalyzed Michael/proton transfer/lactamization (NCMPL) cascade, allowing the total synthesis of α_1a_ adrenergic receptor antagonist (**47**) (Figure 25) [67]. 

Quinolizin-4-ones (**49**) have been prepared from 6-ciano-**3,4-DHPo’s** derivatives (**48**) with low to high yields (20–90%) through allylation/intramolecular Heck reaction sequence (Figure 26). Quinolizin-4-ones showed attractive biological activities related to CNS diseases, including Alzheimer’s [44].

Additionally, in 2019, simple 2-pyridones were applied in synthesizing alkaloids and alkaloid-inspired compounds based on the piperidine or pyridine framework [93], demonstrating the versatility of the revised scaffold.

## 6. Biological Activity

DHPos and its derivatives possess a wide variety of biological activities such as vasorelaxant [57], reverse transcriptase inhibition of human immunodeficiency virus-1 [5,83,94], Rho-kinase inhibitors [12], anticancer [3], antibacterial [4], human rhinovirus 3C protease inhibitors [6,7], urease inhibitors [34], antifungal activity [95], glycine/NMDA receptor inhibitor [8], and as cellular transport [9].

-
**
*Vasorelaxant activity*
**


The regulation of blood pressure depends on vascular tone. In addition, nitric oxide (NO) is an excellent vasodilator molecule, but a low production of endothelium-derived NO causes a diminished vasodilator tone. This increases vascular resistance, which contributes to elevated blood pressure [96]. Therefore, various research groups are focused on finding compounds with vasorelaxant activity. A recent study demonstrated that 3-cyano-pyridin-2-ones (Figure 9) show a significant vasorelaxant. Three of them are the most potent and revealed an endothelium-independent effect.

In addition, 3-cyano-2-pyridone derivatives were synthesized as calcium channel blockers and probable PD3 and PD4 inhibitors, taken as comparison; nifedipine for L-type calcium channel blocker, milrinone and amrinone for PD3 and PD4 inhibition. These compounds also have antihypertensive and vasorelaxant activity [57].

-
**
*HIV-1 inhibitors*
**


Non-nucleoside reverse transcriptase inhibitors (NNRTIs) play an essential role in treating HIV infections. They have been used as the main target in the attack against this virus, and most of them present butterfly-like conformation [94,97]. This conformation facilitates the intramolecular interactions between receptor and ligand. The NNRTI interacts specifically with the HIV reverse transcriptase (RT) substrate-binding site and inhibits its replication. Pyridone derivatives act in this way due to their favorable conformation. Hence, they are highly active against HIV-1. Parreira et al. reported a series of 22 **DHPo** derivatives such as **50** (Figure 10), which are inhibitors of HIV-1 [94]. In 2001, they synthesized 32 **DHPo** and proved their HIV-1 inhibitor activity; one of these compounds (**51**) (Figure 10) showed higher activity [90]. Additionally, another research group described the use of pyridine cocktails to attack viral variants that exhibit drug resistance [98].

-
**
*Antitumor activity*
**


Several **DHPos** have emerged during the last twenty years with potent antitumor activity. They have been mainly tested against P388 lymphocytic leukemia cells, demonstrating potential antitumor activity [99,100]. Oxygen-containing functional groups play an essential role in P388 activity; at least two groups are required. Hwang group reported a series of compounds (**52**) with high activity (Figure 11) [99]. Additionally, 3-Hydroxy-2- pyridone Nucleosides (**53**) [100] and series 2-pyridones (**54**) were reported (Figure 11) [3]. These studies demonstrate the potential utility of 3,4-Dihydro-2(1*H*)-pyridones as building blocks in drug design.

-
**
*Antibacterial and antifungal activity*
**


**DHPos** derivatives present favorable properties as antibacterial agents against multidrug-resistant bacteria such as streptococci and anaerobic microorganisms [4]. In 2018, in vitro antimicrobial activity was reported in some 4-(biphenyl-4-yl)-1,4-dihydropyridine and 4-(biphenyl-4-yl)pyridine derivatives, followed by molecular docking and DFT studies [101]. Ahamed et al. have tested in vitro antibacterial activity of some 1,4-dihydropyridine derivatives against *Escherichia coli*, *Pseudomonas aeruginosa*, *Staphylococcus epidermidis*, *Staphylococcus aureus*, and *Klebsiella pneumoniae*. Most of these compounds were highly active against *E. coli*, and some even showed antifungal activity [102].

-
**
*Other biological activities*
**


Although the **DHPos** have been exceptionally well-explored as a vasorelaxant, they have a privileged scaffold that could act as human rhinovirus 3C protease inhibitors, urease inhibitors, and Rho-Kinase N-methyl-D-aspartate (NMDA) inhibitors. Recent studies have reported a series of 3,4-dihydro-2-pyridone derivatives, from which 4-(4-nitrophenyl)-5-methoxycarbonyl-6-methyl-3,4-dihydropyridone (**3,4-DHPo**) exhibited the most potent activity (IC_50_ = 29.12 μM). This inhibitory activity grows with the increase in the electron-withdrawing ability of the groups [34].

NMDA inhibitors are highly interesting in pharmaceutical research due to their application in treating moderate to severe Alzheimer’s disease [103]. A series of 122 **DHPos** derivatives have been proven to be NMDA inhibitors through QSAR methodology [8]. Five of them, whose structures have a **DHPo** ring fused to a substituted aromatic ring (**55** to **59**), showed the highest inhibitory capacity (Figure 12).

Additionally, some dihydropyridines act as calcium channel blockers, potential candidates for schizophrenia and antihypertensive treatment [104,105].

## 7. Conclusions

Since the first accidental **3,4-DHPo** synthesis, the most extensive method reported to obtain these derivatives proceeded via a multicomponent reaction (MCR-4CR) of Meldrum’s acid, a *β*-keto-ester, and an aromatic aldehyde in the presence of ammonium acetate. This experimental procedure is simple and allows a wealth of molecular diversity depending on substituents in the starting reagents. Besides, this base strategy has been extended to nonconventional methods such as Microwave-, Ultrasound-, or Infrared-assisted reactions, allowing us to increase the efficiency by reducing the reaction time; increasing the yields; and, for some of them, eliminating the reaction solvents, as an important contribution to Green Chemistry. SPOS and LPOS have also been applied, allowing for obtaining a combinatorial library of these structures.

All **3,4-DHPo** synthesized showed a broad range of biological activity highlighted as vasorelaxants and antihypertensives due to its structural similarity with 1,4-DHPs. Many of them also showed activity as antitumors, HIV-inhibitors, or antibacterial and antifungal, among others. On the other hand, **3,4-DHPo** have become excellent synthetic precursors of many different complex structures with exciting applications.

High expectations still surround the next generation and further evolution of **3,4-DHPo**. The improvement of asymmetric synthesis methodologies will allow the enantioselective preparation of this heterocycle, opening up the possibilities of enhancing biological activities. In the short and long term, more complex structures can be synthesized using **DHPo** as base molecular skeletons. In addition, the multicomponent platforms that characterize **DHPo** synthesis will promote the use of virtual screening tools and combinatorial chemistry, allowing us access to leading compounds based on this heterocycle. All described advancements contribute to an in-depth understanding of the potential of this scaffold and pave the way to apply novel 3,4-DHPo-based derivatives for further rational development in drug discovery.

## Data Availability

The data presented in this study are available on request from the corresponding author.

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
