# Peer review of "3,4-Dihydro-2(1*H*)-Pyridones as Building Blocks of Synthetic Relevance"

_molecules, 2022, doi:10.3390/molecules27165070_

Round 1

Reviewer 1 Report

1.- Throughout the document, inconsistencies were found in writing citations.

2.- Do not use the last name and instead, use the numbering. At the end of each paragraph, a period should be placed after the brackets.

3.- Verify the nomenclature of the names of the chemical structures, sometimes they use big letters and sometimes with a subindex, the style should be standardized.

4.- Lacking, an in-depth analysis of the advantages and disadvantages of each method used for the synthesis of 3,4-Dihydro-2(1H)-pyridones. as well as a section on advances and future trends in the field of 3,4-Dihydro-2(1H)-pyridones

5.- review the  file

Author Response

Dear reviewer,

I'm grateful you the opportunity to resubmit a revised version of the manuscript Molecules-1813668 "3,4-Dihydro-2(1H)-pyridones as building blocks of synthetic relevance."

It is a pleasure for us, received all the constructive comments and suggestions which improved our article.

Please, find enclosed a point-by-point response to the reviewer's comments.

I appreciate your suggestions and comments. All have been incorporated in this version of the article (please find enclosed)

We are confident that this manuscript will merit publication in Molecules.

POINT-BY-POINT RESPONSE TO REVIEWER 1

All suggestions and comments have been incorporated in the second version of this paper.

Points to address:

1.- Throughout the document, inconsistencies were found in writing citations.

All inconsistencies were corrected.

2.- Do not use the last name and instead, use the numbering. At the end of each paragraph, a period should be placed after the brackets.

In all cases, the period was placed after the brackets.

3.- Verify the nomenclature of the names of the chemical structures, sometimes they use big letters and sometimes with a subindex, the style should be standardized.

The style was standardized.

4.- Lacking, an in-depth analysis of the advantages and disadvantages of each method used for the synthesis of 3,4-Dihydro-2(1H)-pyridones. as well as a section on advances and future trends in the field of 3,4-Dihydro-2(1H)-pyridones

In page 13, Line 375-383: A paragraph related to advantages and disadvantages of the synthetic methods have been included.

In page 21, Line 578: Further Perspective have been fused to Conclusions.

5.- review the  file

All recommendations and comments mentioned in the file were included in the revised version.

Sincerely yours,

Professor Hortensia María Rodríguez Cabrera

Reviewer 2 Report

My general impression of this review is that it is incomplete, missing several articles on the synthesis 3,4-Dihydro-2(1H)-pyridones of in easily obtained journals such as that of Stojanovic, J. Org. Chem. 2020, 85, 21, 13495-13507, https://pubs.acs.org/doi/full/10.1021/acs.joc.0c01537, Hakimi, Res Pharm Sci. 2017 Oct; 12(5): 353-363, doi: 10.4103/1735-5362.213980, and others found on a fairly cursory Google search of the scaffoold name. However, Suarez-Navarro self cites in nearly one quarter of the over 100 references and many of those are crystallographic papers which are deposited in what I would describe as a minimum publishable unit. The content of the review is OK, but incomplete. There are numerous problems with wording (many, many run on sentences). Some of that can be attributed to non-native English and can be remedied with editorial direction. However, the proof should have been reviewed better by the authors with several broken link messages in the text. Based on these issues, I would recommend to reconsider after major revision.

Author Response

Dear colleague,

I'm grateful you the opportunity to resubmit a revised version of the manuscript Molecules-1813668 "3,4-Dihydro-2(1H)-pyridones as building blocks of synthetic relevance."

It is a pleasure for us, received all the constructive comments and suggestions which improved our article.

Please, find enclosed a point-by-point response to the reviewer's comments.

I appreciate your suggestions and comments. All have been incorporated in this version of the article (please find enclosed)

We are confident that this manuscript will merit publication in Molecules.

POINT-BY-POINT RESPONSE TO REVIEWER 2

It is a pleasure for us, received all the constructive comments and suggestions which improved our article.

Points to address:

My general impression of this review is that it is incomplete, missing several articles on the synthesis 3,4-Dihydro-2(1H)-pyridones of in easily obtained journals such as that of Stojanovic, J. Org. Chem. 2020, 85, 21, 13495-13507, https://pubs.acs.org/doi/full/10.1021/acs.joc.0c01537, Hakimi, Res Pharm Sci. 2017 Oct; 12(5): 353-363, doi: 10.4103/1735-5362.213980, and others found on a fairly cursory Google search of the scaffoold name. However, Suarez-Navarro self cites in nearly one quarter of the over 100 references and many of those are crystallographic papers which are deposited in what I would describe as a minimum publishable unit. The content of the review is OK, but incomplete.

It is important to note that the present review is focused on 3,4-DHPo (See below, A). As the reviewer highlight, there is a broad range of similar scaffolds with different related positions for carbonyl and unsaturation. In this regard, the first reference (J. Org. Chem. 2020, 85, 21, 13495-13507, https://pubs.acs.org/doi/full/10.1021/acs.joc.0c01537) is related to a different scaffold (See below, B).

The other mentioned reference (Res Pharm Sci. 2017 Oct; 12(5): 353-363, doi: 10.4103/1735-5362.213980) does appear included in the first version of the document as 33.

Following the review suggestion, a new search was carried out so that no critical reference is left unmentioned.

There are numerous problems with wording (many, many run on sentences). Some of that can be attributed to non-native English and can be remedied with editorial direction. However, the proof should have been reviewed better by the authors with several broken link messages in the text.

The newly submitted version was exhaustively revised to improve the wording and general grammatical issues.

Sincerely yours,

Professor Hortensia Rodríguez

Reviewer 3 Report

The article 3,4-Dihydro-2(1H)-pyridones as building blocks of synthetic 2 relevance after consideration of major comments.

The study is useful for medicinal chemist and drug designer for synthesis of main core of biological active drugs

.

1)      Abstract authors should contain conclusion.

2)      Introduction, the rational of this study.

3)      Authors should add part to this study to compare between the different synthetic methods.

4)      The conclusion should be improved.

5)      Page 3, no.98, what it is mean

6)      Scheme 5, authors should mention that pyridine used as catalyst and ethanol an solvent.

7)      It is better to add table for comparing methods condition, reagents and yield.

8)      It may be better to remove scheme 9 as it contains fused ring synthesis.

Author Response

Dear colleague,

I'm grateful you the opportunity to resubmit a revised version of the manuscript Molecules-1813668 "3,4-Dihydro-2(1H)-pyridones as building blocks of synthetic relevance."

It is a pleasure for us, received all the constructive comments and suggestions which improved our article.

Please, find enclosed a point-by-point response to the reviewer's comments.

I appreciate your suggestions and comments. All have been incorporated in this version of the article (please find enclosed)

We are confident that this manuscript will merit publication in Molecules.

POINT-BY-POINT RESPONSE TO REVIEWER 3

The article 3,4-Dihydro-2(1H)-pyridones as building blocks of synthetic 2 relevance after consideration of major comments.

The study is useful for medicinal chemist and drug designer for synthesis of main core of biological active drugs

  • Abstract authors should contain conclusion.

Following the reviewer suggestion, the abstract was modified. See Page 1 Line 27 to 32.

  • Introduction, the rational of this study.

The more specific objective of this review has been rewritten. See Page 2 Line 68 to Page 3 Line 73.

  • Authors should add part to this study to compare between the different synthetic methods.

In page 13, Line 375-383: A paragraph related to advantages and disadvantages of the synthetic methods have been included.

  • The conclusion should be improved.

The conclusion was improved. See Page 21 Line 586 to 606

  • Page 3, no.98, what it is mean

There are some problems with the references in the document. All were corrected in this new version of the manuscript.

  • Scheme 5, authors should mention that pyridine used as catalyst and ethanol an solvent.

The reviewer suggestion was included in the text (Page 4, Line 128) and the Scheme 4 was also modified (See Scheme 4)

  • It is better to add table for comparing methods condition, reagents and yield.

Not a table, but in page 13, Line 375-383: A paragraph related to advantages and disadvantages of the synthetic methods have been included.

  • It may be better to remove scheme 9 as it contains fused ring synthesis.

Following the reviewer suggestion, the scheme 9 was eliminated.

 Sincerely yours

Professor Hortensia Rodríguez

Reviewer 4 Report

This manuscript submitted by Rodriguez et al. described 3,4-Dihydro-2(1H)-pyridones as building blocks of synthetic relevance and application in biological science. The review article is well written, but the authors did not plan properly, such as using year-by-year references, and they did not focus properly on how in the last ten to fifteen years' researchers have taken the lead for the synthesis of 3,4-Dihydro-2(1H)-pyridones using various synthetic tools and applications in various field (Biological science).

However, the authors tried to cover the majority of the section but were unable to cover all of the literature support in detail to make this review article more interesting for readers. Therefore, this review in its present form is not recommended for publication.

Author have not explained several research papers such as:

ACS Comb. Sci. 2011, 13, 4, 421–426, Ultrasonic Sonochemistry, 2011, 18, 32-36, Chem. Commun., 2013, 49, 4346, Res Pharm Sci. 2017 Oct; 12(5): 353–363, J. Org. Chem. 2000, 65, 26, 9103–9113, Org. Biomol. Chem., 2017, 15, 5171, Int. J. Mol. Sci. 201112(4), 2641-2649, Chemical Physics Letters 649 (2016) 84–87, Synthesis 2019; 51(18): 3369-3396.

Author Response

Dear colleague,

I'm grateful you the opportunity to resubmit a revised version of the manuscript Molecules-1813668 "3,4-Dihydro-2(1H)-pyridones as building blocks of synthetic relevance."

It is a pleasure for us, received all the constructive comments and suggestions which improved our article.

Please, find enclosed a point-by-point response to the reviewer's comments.

I appreciate your suggestions and comments. All have been incorporated in this version of the article (please find enclosed)

We are confident that this manuscript will merit publication in Molecules.

POINT-BY-POINT RESPONSE TO REVIEWER 4

This manuscript submitted by Rodriguez et al. described 3,4-Dihydro-2(1H)-pyridones as building blocks of synthetic relevance and application in biological science. The review article is well written, but the authors did not plan properly, such as using year-by-year references, and they did not focus properly on how in the last ten to fifteen years' researchers have taken the lead for the synthesis of 3,4-Dihydro-2(1H)-pyridones using various synthetic tools and applications in various field (Biological science).

The review focuse on cover nomenclature, synthesis, and biological activity of 3,4-DHPo; but also highlights the importance of this  heterocycle as building blocks of other relevant structures.  

However, the authors tried to cover the majority of the section but were unable to cover all of the literature support in detail to make this review article more interesting for readers. Therefore, this review in its present form is not recommended for publication.Author have not explained several research papers such as:ACS Comb. Sci. 2011, 13, 4, 421–426, Ultrasonic Sonochemistry, 2011, 18, 32-36, Chem. Commun., 2013, 49, 4346,Res Pharm Sci. 2017 Oct; 12(5): 353–363, J. Org. Chem. 2000, 65, 26, 9103–9113, Org. Biomol. Chem., 2017, 15, 5171, Int. J. Mol. Sci. 201112(4), 2641-2649, Chemical Physics Letters 649 (2016) 84–87, Synthesis 2019; 51(18): 3369-3396.

The mentioned papers were included and explained in the new version of this manuscript, except J. Org. Chem. 2000, 65, 26, 9103–9113, which was not considered relevant.

Sincerely yours,

Professor Hortensia Rodríguez

Round 2

Reviewer 1 Report

Line 602, Hwang and Driscoll, does not have the format.Line 638, within the CONCLUSIONS section, Future perspective, it is necessary to delve a little deeper into the perspectives, what are the empty spaces in the 3,4-DHPo knowledge?, what is expected in the short and long term?. The section, 7. Conclusions and Future Perscpective, it is not necessary to put the words together, leave only 7. Conclusions, within the text make reference to future perspective. just need to order this section.

Author Response

To:       Reviewer

           Molecules

           MDPI publication

Dear colleague,

I'm grateful you the opportunity to resubmit a revised version of the manuscript Molecules-1813668 "3,4-Dihydro-2(1H)-pyridones as building blocks of synthetic relevance."

Thanks again for all you constructive comments and suggestions which improved our article.

Please, find enclosed a point-by-point response to your comments.

I appreciate your suggestions and comments. All have been incorporated in this revised version of the article (please find enclosed)

We are confident that this manuscript will merit publication in Molecules.

Sincerely yours,

Hortensia María Rodríguez Cabrera, PhD

Professor

School of Chemical Science and Engineering

e-mail: hmrodriguez@yachaytech.edu.ec

Tf: 0994336513

POINT-BY-POINT RESPONSE TO REVIEWER 1

Line 602, Hwang and Driscoll, does not have the format.Line 638, within the CONCLUSIONS section, Future perspective, it is necessary to delve a little deeper into the perspectives, what are the empty spaces in the 3,4-DHPo knowledge?, what is expected in the short and long term?. The section, 7. Conclusions and Future Perscpective, it is not necessary to put the words together, leave only 7. Conclusions, within the text make reference to future perspective. just need to order this section.

Points to address:

1.- Line 602, Hwang and Driscoll, does not have the format

Done the correction

2.- Line 638, within the CONCLUSIONS section, Future perspective, it is necessary to delve a little deeper into the perspectives, what are the empty spaces in the 3,4-DHPo knowledge?, what is expected in the short and long term?.

Agree, a couple of sentences were included to deeper in the perspective in the short and long term. (See Lines 602-605)

3.- The section, 7. Conclusions and Future Perscpective, it is not necessary to put the words together, leave only 7. Conclusions, within the text make reference to future perspective. just need to order this section.

Done.

Reviewer 3 Report

the authors reply to all comments 

the article is accepted in the present form

Author Response

Dear reviewer,

thanks for accepting the manuscript in the present form.

I appreciate all your suggestions and comments. 

Reviewer 4 Report

Manuscript will be accepted in current form 

Author Response

(The authors gave the same response as above.)
